# Reusability of P3 Facial Filter in a Pandemic Emergency: A 3D Analysis of Filter Microstructure with X-ray Microtomography Images after Dry Heat and UV Sterilization Procedures

**DOI:** 10.3390/ijerph19063435

**Published:** 2022-03-14

**Authors:** Luca Borro, Massimiliano Raponi, Andrea Del Fattore, Franco Zanini, Francesca di Lillo, Adriano Contillo, Veronica Bordonaro, Eleonora Di Piazza, Alberto E. Tozzi, Aurelio Secinaro

**Affiliations:** 1Advanced Cardiothoracic Imaging Unit, Department of Imaging, Bambino Gesù Children’s Hospital, IRCCS, 00165 Rome, Italy; veronica.bordonaro@opbg.net (V.B.); aurelio.secinaro@opbg.net (A.S.); 2Medical Direction, Bambino Gesù Children’s Hospital, IRCCS, 00165 Rome, Italy; massimiliano.raponi@opbg.net; 3Bone Physiopathology Research Unit, Genetics and Rare Diseases Research Area, Bambino Gesù Children’s Hospital, IRCCS, 00165 Rome, Italy; andrea.delfattore@opbg.net; 4Elettra-Sincrotrone Trieste, 34149 Trieste, Italy; franco.zanini@elettra.eu (F.Z.); francesca.dilillo@elettra.eu (F.d.L.); adriano.contillo@elettra.eu (A.C.); 5UOC Endocrinologia e Diabetologia, Dipartimento di Scienze Mediche e Chirurgiche, Fondazione Policlinico Universitario A. Gemelli–IRCCS, 00168 Rome, Italy; eleodp@gmail.com; 6Multifactorial and Complex Diseases Research Area, Bambino Gesù Children’s Hospital, 00165 Rome, Italy; albertoeugenio.tozzi@opbg.net

**Keywords:** COVID-19, filtering facepiece respirator, facial protection mask, reuse of facial mask

## Abstract

Objective: Our goal is to evaluate the effects of heat and ultraviolet (UV) irradiation on P3 facial respirator microstructure. Intervention: P3 facial filters were exposed to dry heat and UV sterilization procedures. Methods: P3 facial filter samples underwent a standardized sterilization process based on dry heat and UV irradiation techniques. We analyzed critical parameters of internal microstructure, such as fiber thickness and porosity, before and after sterilization, using 3D data obtained with synchrotron radiation-based X-ray computed microtomography (micro-CT). The analyzed filter has two inner layers called the “finer” and “coarser” layers. The “finer” layer consists of a dense fiber network, while the “coarser” layer has a less compact fiber network. Results: Analysis of 3D images showed no statistically significant differences between the P3 filter of the controls and the dry heat/UV sterilized samples. In particular, averages fiber thickness in the finer layer of the control and the 60° dry heated and UV-irradiated sample groups was almost identical. Average fiber thickness for the coarser layer of the control and the 60° dry heated and UV-irradiated sample groups was very similar, measuring 19.33 µm (±0.47), 18.33 µm (±0.47), and 18.66 µm (±0.47), respectively. There was no substantial difference in maximum fiber thickness in the finer layers and coarser layers. For the control group samples, maximum thickness was on average 11.43 µm (±1.24) in the finer layer and 59.33 µm (±6.79) in the coarser layer. Similarly, the 60° dry heated group samples were thickened 12.2 µm (±0.21) in the finer layer and 57.33 µm (±1.24) in the coarser layer, while for the UV-irradiated group, the mean max thickness was 12.23 µm (±0.90) in the finer layer and 58.00 µm (±6.68) in the coarser layer. Theoretical porosity analysis resulted in 74% and 88% for the finer and coarser layers. The finer layers’ theoretical porosity tended to decrease in dry heat and UV-irradiated samples compared with the respective control samples. Conclusions: Dry heat and UV sterilization processes do not substantially alter the morphometry of the P3 filter samples’ internal microstructure, as studied with micro-CT. The current study suggests that safe P3 filter facepiece reusability is theoretically feasible and should be further investigated.

## 1. Introduction

In early 2020s, the coronavirus pandemic involved the entire world, and the use of individual protective devices for protection from SARS-CoV-2 infection has been forcefully highlighted.

SARS-CoV-2 is an RNA virus first isolated in the Wuhan (China) in December 2019 and is responsible for COVID-19 disease, which causes severe interstitial pneumonia [1]. The first way of diffusion of SARS-CoV-2 is through saliva droplets with a diameter ranging from 1 µm to 500 µm [2] ejected by sneeze, cough, and speech. Another way of viral diffusion is by contact via contaminated surfaces [3]; indeed, studies have shown that SARS-CoV-2 remains on contaminated surfaces for several hours, depending on the type of material considered [4]. The global pandemic has caused more than 412 million cases worldwide, of which more than 5.82 million have died as of February 2022 due to severe lung infections caused by pulmonary interstitial disease and lungs parenchymal consolidations.

The spread of the Omicron variant of SARS-CoV-2, for the first time sequenced in South Africa starting in November 2021, has led many countries to recommend the use of P2 face masks instead of surgical masks in consideration of the increased contagiousness of Omicron compared to the Delta variant.

Personal protective equipment is crucial to containing the spread of contagion worldwide in the absence of practical therapeutic approaches [5]. Previous works have highlighted the effectiveness of facial protection masks to protect against viral pathogens [6,7]. Based on scientific evidence, governments worldwide have strongly recommended using facial protection masks and respirators for the general population and health professionals.

Different protective devices are used by the general population and in health contexts. These devices can be divided into different categories according to their function and material, including cloth masks, surgical masks, and filtering facepiece respirators.

The Filtering Facepiece Respirator (FFR) is a high-protection face mask divided into different types depending on the design and the purpose of use. The nomenclature of FFR varies in other countries. FFP2 and FFP3 (UK designation) respirators correspond to N95 and N99 (USA designation) respirators. In Europe, FFP respirators must meet the European Standard EN149:2001 [4]. The most frequently used FFR in hospitals are FFP2 (N95) and FFP3 (N99). The efficiency of FFP3 respirators is higher than FFP2 respirators, with penetration of polydisperse NaCl aerosols particles through the filter being <1% for FFP2 and <0.03% for FFP3 [8].

FFRs consist of non-woven polypropylene fibrous filter media, which capture microscopic particles. Several filtration mechanisms typical of facial respirators characterize the interaction between the porous media’s fibers and the motion of particles passing through the filter. These mechanisms are classified into four different types: interception, impaction, diffusion, and electrostatic attraction [9].

Smaller particles (with a diameter of <0.6 µm) that move at low speed within the filter are primarily blocked by impaction and interception. With impaction, the particles that enter the filter are blocked by the fibers’ impact caused by the particle’s inertia. With the interception mechanism, the particles passing close to the fiber follow their streamline. Particles less than 0.1 µm in diameter, on the other hand, are mainly captured by the diffusion mechanism, which is caused by Brownian air motions that divert a particle’s motion until it comes into contact with a fiber. Electrostatic attraction requires that the fibers be electrically charged to determine an electrostatic attraction with the particles in motion [10].

Massive mask demand during SARS-CoV-2 pandemic has led to a sharp decrease in the supply of respirators on the market. This caused an increased risk of contagion in the early stages of the pandemic, especially for the health workers fighting the pandemic in intensive care units around the world.

For these reasons, it is relevant to understand the practical reusability of FFRs to enable adequate protection from contagion diffusion for all health professionals and for the general population.

Recent studies have also found infecting viral particles on facial protective masks for up to 4 days in the innermost layers and 7 days in the outermost layers [11]. Several studies have confirmed that the coronaviruses are stable at different temperatures: at 4 degrees Celsius (°C), at room temperature (20 °C), and at 37 °C. However, these viruses become non-infectious after 90 min at 56 °C, 60 min at 67 °C, and 30 min at 75 °C [12]. It was also shown that UV radiation for 60 min decreases the viral load to negligible levels [11].

There have been few decontamination studies carried out using dry heat. A filter sample was placed at 70 °C in dry heat for 10, 20, 30, and 60 min in a recently published study. These contamination cycles caused light drops in filtration performance; however, the reduction of filtration performance revealed with this method is lower than that observed by a 70% ethanol sterilization cycle, which led to filter structure alterations. The decontamination process carried out with UV rays, on the other hand, did not result in any significant structural alterations of the filter [13].

The studies analyzed so far have used highly professional sterilization systems, and it would be particularly challenging to use them in the emergency context due to the SARS-CoV-2 pandemic, which would require continuous reuse of respirators otherwise not commercially available. These studies also focused more on UV decontamination and less on dry heat sterilization. Despite some reports showing a decrease of filtering capacity following filter sterilization treatments, none of the studies conducted a morphometric analysis of the micro-architecture of facial respirators to check the integrity of polypropylene fibers post decontaminating treatment.

This study evaluated the microstructure of the fibrous porous medium of a P3 filter sample (Model 5935–3M, 3M Italy, Milan, Italy) before and after a filter sterilization procedure via dry heat and UV radiation.

The study aims to test whether the sterilization methods used cause alterations in the polypropylene filter fibers’ structure.

## 2. Materials and Methods

### 2.1. Samples

Nine P3 particulate filters (Mod. 5935–3M, 3M Italy, Milan, Italy) were used. This filter is commonly used as protection against particulates, dust, fumes, and especially against spray droplets, and it is compliant with AS/NZS 1716:2012 regulations. Given the scarcity of facemasks in during the coronavirus pandemic for the general population, we decided to use only one filter for our study. We made subsamples to conduct our experiment without depriving the market of essential protection devices, especially for health workers. The dimensions of the entire filter are 105 mm (Height) × 85 mm (Max Width) × ~4 mm (Thickness). The dimensions of our filter samples are: 10 × 10 mm.

### 2.2. Sterilization

Heat sterilization was performed incubating the samples at 60 °C for 40 min using the Medite^®^ TDO Sahara oven (Medite, Medite, Burgdof, Germany). Regarding the UV irradiation, samples were subjected to UV for 120 min using the G15T8 lamp (Sankio Denki, Sankyo Denki, Kanagawa, Japan).

### 2.3. Synchrotron Radiation X-ray Microtomography

The synchrotron radiation-based X-ray computed microtomography was performed at the SYRMEP beamline of the Italian Synchrotron Radiation facility (Elettra, Trieste, Italy) [14] with a polychromatic X-ray beam (mean energy of about 17 keV) and a sCMOS detector (Orca Flash, Hamamatsu Photonic K.K., Iwata, Japan) optically coupled with a 17-μm thick GGG (Gd3Ga5O12:Eu) scintillator. For each sample, 1800 tomographic projections were acquired over 180 degrees, with a single exposure time of 50 ms, in propagation-based phase-contrast imaging mode [15]. A phase retrieval algorithm as in Paganin [16] was applied before the tomographic reconstruction, which was performed through GPU-based filtered back-projection using the SyrmepTomoProject software (Elettra Sincrotrone, Trieste, Italy) [17]. The reconstructed pixel size was 2 × 2 × 2 μm^3^.

### 2.4. Image Processing

All image analyses were conducted using Fiji software version 1.52p (NIH, Bethesda, MD, USA). Fiji is an updated version of ImageJ, an open source image processing program designed for multidimensional scientific images. It is highly extensible, with several plugins and scripts for performing various tasks and parameter measurements on 2D or 3D images. After an initial analysis of 3D volumes from the X-ray microtomography of the filter, it was immediately evident that the micro-architecture of the filter’s polypropylene fibers could be defined as a “bone-like” structure, consisting mainly of random-oriented rods in space alternating with empty spaces. For this reason, BoneJ, a free open source ImageJ plugin often used for bone analysis, was used. In order to deeply investigate the microstructure of sample fibers, an in-depth analysis was conducted using the thickness and volume fraction tools of the Fiji BoneJ plugin [18]. The thickness tool allows the detection of fibers’ mean thickness in a fibrous porous media such as a facepiece respirator filter, while the volume fraction allows the analysis of the ratio between void space and filled space occupied by fibers in the filter.

### 2.5. Local Thickness Measurement

Given a domain and a generic point within it (Figure 1), the local thickness is calculated as the diameter of the largest sphere that contains the point and that is wholly contained within the structure we want to measure:(1)τ(p)=2⋅max({r|p ϵ sph(x,r)⊆Ω,x∈Ω})

Equation (1): Local Thickness [τ(*p*)] calculating equation, where sph(x,r) is the points inside the sphere with center x and radius r. Since local thickness must be defined for all points contained within the considered structure, it is necessary to define a volume-based local thickness on the entire structure through an integration:(2)$$\bar{\tau }\;=\;\frac{1}{\mathrm{vol}\left(\Omega \right)}{\;\int \int \int \;}_{\Omega }\tau \left(x\right)d^{3}x\;\mathrm{Vol}\left(\Omega \right)={\;\int \int \int \;}_{\Omega }d^{3}x$$

Equation (2): Equation for calculating the local thickness ($\bar{\tau }$) for each point contained within the considered volume (Ω).

The maximum local thickness is represented by the largest sphere that can be contained within the structure:(3)$$\tau_{\max}=\max\;\left(\;\left\{\tau \left(p\right)|\;p\;\epsilon \;\Omega \;\right\}\right)$$

Equation (3): The maximum local thickness ($\tau_{\max}$) is equivalent to the largest sphere contained inside the considered volume.

The thickness tool requires an 8-bit binarized image obtained with the semi-automatic segmentation based on a thresholding algorithm in Fiji software.

For the finer layer, segmentation was done with the thresholding algorithm in “mean” mode. However, thresholding in the “default” mode was used for the coarser layer.

The mean modality uses the mean of grey levels as the threshold cutoff. The default modality is a variation of the IsoData mode [19] that divides the image in an object and background based on an initial threshold. All the averages of the pixels at or below the initial threshold and the pixels above are computed. The averages of those two values are calculated, and the process is repeated until the threshold is larger than the composite average.

### 2.6. Theoretical Filter Porosity

The theoretical filter porosity is the measure of the voids inside the porous media; it is defined as the ratio of void volume to the total volume of the sample and is expressed as a percentage. Porosity is defined as the ratio:(4)$$\phi =\frac{v_{v}}{v_{t}}$$

Equation (4): The ratio between the void volume (V_v_) and the total volume (V_t_) represents the theoretical porosity of the filter (ϕ), where V_v_ is the volume of void space and V_t_ is the total volume of the considered material. For the facemask filters, the voids in the porous media contain air. One of the methods for calculating the material’s porosity is the X-ray microtomography, which allows 3D reconstruction of both the void space and the full spaces of the sample. The porosity analysis is based on each of the nine samples analyzed through the Fiji software using BoneJ’s volume fraction function.

### 2.7. Statistics

Statistical analysis was performed by one-way analysis of variance. Results were analyzed using the GraphPad Prism software version number 5 (GraphPad Software, San Diego, CA, USA), and they were expressed as mean ± standard deviation (SD). A result of *p* < 0.05 was considered statistically significant.

## 3. Results

The internal structure of the P3 filter (5935-3M) consists of two main layers: a finer layer and a coarser layer (Figure 2) with 1.72 mm and 1.18 mm of thickness, respectively. The difference between these two layers is that the finer layer consists of more compact and “rod-like” fibers with a roughly cylindrical structure, while the coarser layer has thinning “plate-like” fibers with narrow rectangular section (Figure 3).

To avoid data alteration of the thickness measurements due to the layers’ inhomogeneity, we analyzed the finer and the coarser texture of layers separately. Several image analysis techniques were used to evaluate the fibrous micro-architecture of the samples. The parameters chosen for the filters’ structural characterization were thickness and porosity of the nano-fibers, which quantify alterations of the fibrous microstructure in the filter samples following the sterilization processes.

### Thickness of Nano-Fibers

The microtomography datasets were analyzed with the BoneJ thickness tool to determine the mean thickness of the polypropylene nano-fibers. The colorimetric map of the fiber thickness did not alter the fiber’s dimensions in the three experimental groups (Figure 4). The image analysis confirmed that the sterilization procedure conducted on the samples did not result in any significant changes in the thickness size of the fibers, measuring the two layers of the filters (finer and coarser layers) (Figure 5A,B). Moreover, heat and UV treatments did not alter the average maximum of fiber thickness in the layers (Figure 5C,D).

The filter samples’ theoretical porosity analysis showed no statistically significant differences in the microstructure of the polypropylene fibers after sterilization treatments (Figure 5E,F).

## 4. Limits of the Study

An important point to be considered to complete the study of filter microstructure alterations after the sterilization process is the fibers’ electrostatic charge. In this study, we did not evaluate if charge remained unchanged during and after the sterilization processes. Future investigation is needed to assess post-sterilization filters’ permeability with laboratory-specific tests that were not the subject of this work.

This study also looked at a single type of P3 facial filter for facial semi-masks and not directly at an FFP3 facial protection mask, due to the scarcity of these devices for research at the moment of analysis. The number of samples analyzed was also small. Although we have not detected significant alterations in the microstructure of the various samples analyzed, a large-scale study of samples could lead to different results.

Unlike UV rays, the 60 °C dry heat sterilization process used in our study is not reported in the literature as a widely used decontamination method, mainly because it takes longer to obtain proper decontamination of the devices [20]. Studies have also shown that dry heat may not be enough to inactivate certain human viruses completely [21] and this is a limitation of our research that can be overcome in the future by using a more conventional sterilization process, such as warm moist heat or microwave-generated steam [20]. However, it was important to establish that sterilization systems that are not particularly complex and generally available within hospitals, such as dry heat and UV irradiation, do not result in substantial changes to the analyzed filters’ microstructure. Based on the encouraging results obtained in this study, we believe that more extensive studies are needed to better understand whether sterilization and reuse of facial protection filters can become a possible and standardized practice within hospitals during a pandemic.

## 5. Conclusions

The current study shows that P3 facial filter exposure to a sterilization process with dry heat 60 °C and UV irradiation does not substantially alter the filter microstructure.

Scientific studies have investigated FFR mask reusability during a pandemic over the years. Several sterilization methods have been evaluated, such as UV radiation, sterilization in ethylene oxide, vaporization of hydrogen peroxide, microwave oven irradiation, and bleach [20]. Based on the results obtained from previous works [22,23], in which the sterilization processes led to an alteration of the filter structure and filtering capacity, we decided to deepen the understanding of this problem with an in-depth analysis of the relationship between the sterilization process and the structural integrity of facial protection filters.

Our study analyzed personal protective devices, investigating the potential for reusability considering that infecting viral particles on facial filters may be found for up to 7 days [13]. Unlike previously cited works that mainly used complex sterilization systems such as moist heat and microwaves, we applied more accessible decontamination processes such as dry heat and UV rays.

The use of X-ray micro-CT allowed us to analyze the three-dimensional microstructure of the devices, thus obtaining accurate volumetric images of the filters analyzed on different spatial planes.

Some other studies have already used X-ray micro-CT technology to study the porous filter microstructure [24], as well as scanning electron microscopy (SEM) [25].

This technique allowed us to analyze the fibrous structure of the filter in detail with a resolution 2 µm, and better understand its geometry and morphometry details to detect any structural changes following the examined sterilization processes. From the analysis of the different filter samples, the sterilization processes did not statistically change the micro-architecture of the filters, which showed no particular alterations of porosity compared to the average porosity values of similar filters reported in previous studies [25,26]. Similarly, the two planned sterilization processes did not significantly reduce the average diameter of the fibers that make up the filter. As polypropylene is a material with a melting point around 160 °C, it is particularly resistant to lower temperatures even for extended periods (60 °C for 40 min), as demonstrated by this work.

## Figures and Tables

**Figure 1 ijerph-19-03435-f001:**
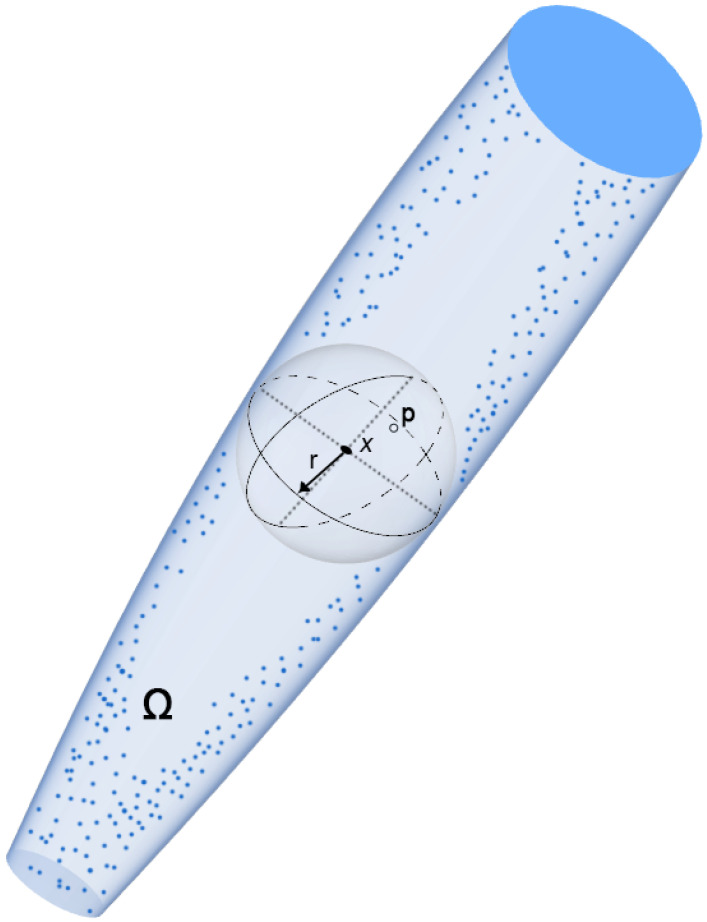
Schematic representation of the local thickness calculation method. The local thickness is calculated for each point P of a generic volume Ω as the diameter of the largest sphere inscribed in the volume that contains the point considered.

**Figure 2 ijerph-19-03435-f002:**
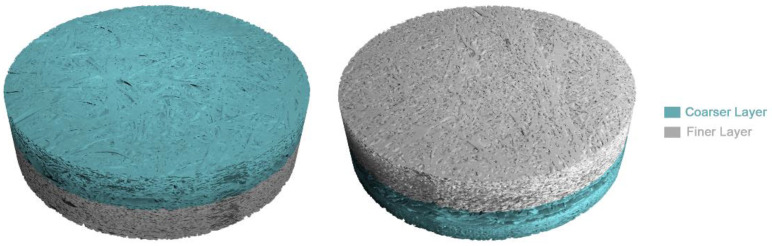
3D reconstructions from micro-CT images with coarser (light green) and finer layers (light grey).

**Figure 3 ijerph-19-03435-f003:**
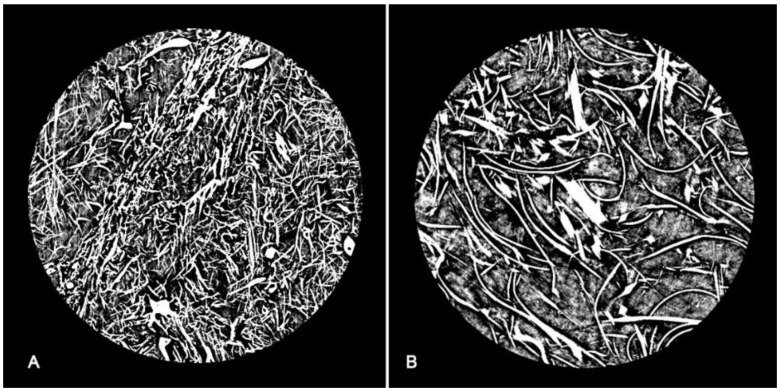
Axial section of the P3 filter from X-ray micro-CT images: (**A**) fine layer consisting of polypropylene fibers with a predominantly cylindrical section (rod-like); (**B**) coarser layer consisting of fibers with a thin, predominantly rectangular section (plate-like), with a greater rarefaction of the fiber distribution.

**Figure 4 ijerph-19-03435-f004:**
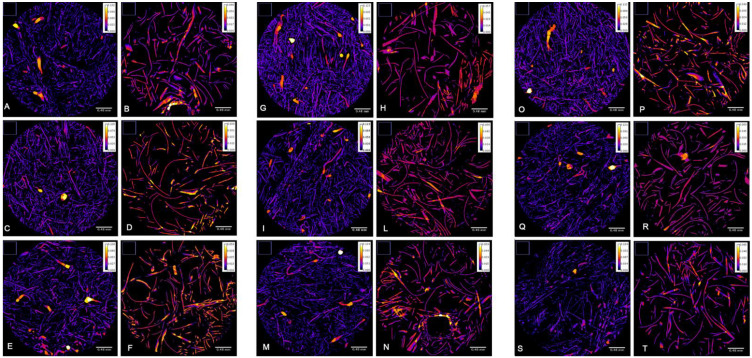
Colorimetric map of the fiber thickness in filter samples. The first two columns represent (**A**,**B**) the thickness of the finer and coarser layers of control Sample 1, (**C**,**D**) control Sample 2, and (**E**,**F**) control Sample 3. The second panel represent the thickness of the finer and coarser layers of (**G**,**H**) 60 °C-treated Sample 1, (**I**,**L**) 60 °C-treated Sample 2, and (**M**,**N**) 60 °C-treated Sample 3. The third panel represents the thickness of the finer and coarser layers of (**O**,**P**) UV-irradiated Sample 1, (**Q**,**R**) UV-irradiated Sample 2, and (**S**,**T**) UV-irradiated Sample 3.

**Figure 5 ijerph-19-03435-f005:**
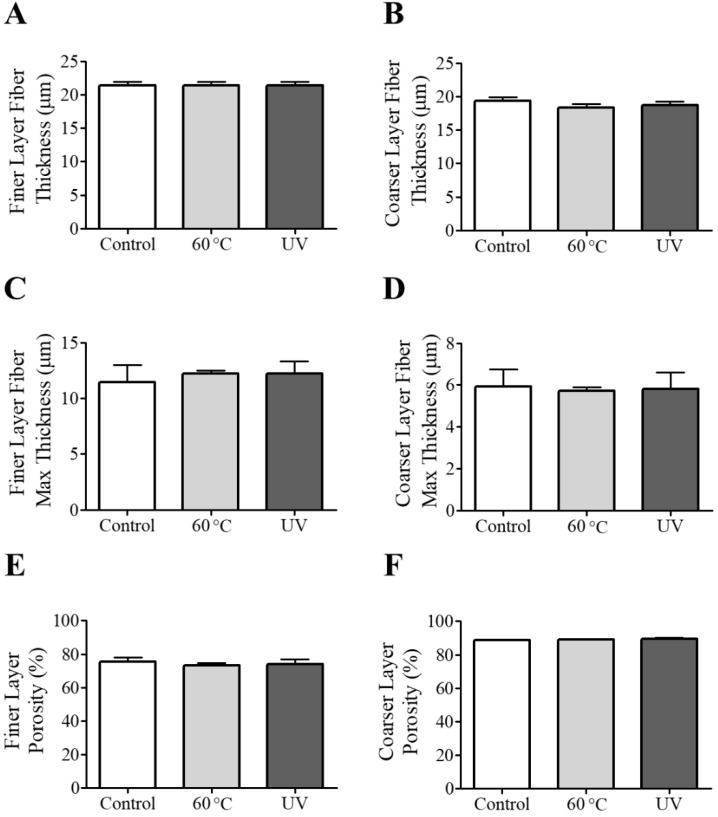
(**A**,**B**) The finer and coarser layer thickness, (**C**,**D**) max thickness and (**E**,**F**) porosity for different types of samples. Results are expressed as mean ± standard deviation.

## Data Availability

The data presented in this study are available on request from the corresponding author.

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
