# Peer review of "Reusability of P3 Facial Filter in a Pandemic Emergency: A 3D Analysis of Filter Microstructure with X-ray Microtomography Images after Dry Heat and UV Sterilization Procedures"

_ijerph, 2022, doi:10.3390/ijerph19063435_

Round 1

Reviewer 1 Report

Revision of the manuscript ijerph-1636458

The paper aim at investigating the effects of UV and dry heat treatments on the microstructure of P3 facial filters in order to see if these sterilization treatments affect the filtration properties of the filter. The study was motivated by the paucity of such devices during pandemic emergencies and therefore by the need to reuse them, which is also a good practice for a sustainable future. The results, based on X-ray computed microtomography and statistical analysis, show that UV and dry heat treatments, while effective decontamination methods, also preserve the microstructure of the filter and therefore its filtrations properties. These results may be helpful to direct sterilization strategies of face musk filters in future applications, and therefore the paper worthy of publication. Minor changes are suggested below.  

Line 88: extra space

Line 88-89: “In the interception…” the phrase appears incomplete

Line 150: “à la Paganin” is this correct?

Line 168-173: there are apparently repetitions, please rephrase

Line 269: “J/cm2” superscript

Line 262-302: there are substantial repetitions of what already said in the introduction and/or experimental sections

Author Response

Dear Reviewer,

thank you very much for your appreciated review.

Below I will reply point by point to your instructions indicating the changes made.

Line 88: extra space
Corrected

Line 88-89: “In the interception…” the phrase appears incomplete
Corrected

Line 150: “à la Paganin” is this correct?
The term is correct and refers to the user algorithm in reference (16).
It is a term used in the field of advanced image analysis.

Line 168-173: there are apparently repetitions, please rephrase
Corrected

Line 269: “J/cm2” superscript
Corrected

Line 262-302: there are substantial repetitions of what already said in the introduction and/or experimental sections
I have rephrased a few sentences from the Conclusions.

Thank you for your support.

Best Regards
Luca Borro

Reviewer 2 Report

The presented topic “Reusability of P3 Facial Filter In a Pandemic Emergency: A 3D Analysis of a Filter Microstructure With X-Ray Microtomography Images After Dry Heat and Uv Sterilization Procedures ” is interesting and it concerns a very important area of COVID-19. This research was well designed and organized. The results can attract a great attention of community. Therefore, I suggest to publish this research after a minor revision as follow:

1) The format of key words is not uniform. Some words are capitalized, some are lowercase.

2) Recheck the referencing format

3) What's the source of P3 Particulate Filter samples??

4) Write the version of the software used: Fiji software

5) write the equation numbers and describe it in text

6) Justify the statement: At higher doses, the respirator's fibrous layers' hardness suffers a substantial loss of resistance, in some cases of more than 90%.

Author Response

Dear Reviewer,

thank you very much for your appreciated review.

Below I will reply point by point to your instructions indicating the changes made.

1) The format of key words is not uniform. Some words are capitalized, some are lowercase.
I tried to correct all the discordant lowercase and uppercase letters.

2) Recheck the referencing format
I have revised the referencing format.

3) What's the source of P3 Particulate Filter samples??
I added the source of the filters. Line 124

4) Write the version of the software used: Fiji software
I added Fiji version. Line 154

5) write the equation numbers and describe it in text
Done.

6) Justify the statement: At higher doses, the respirator's fibrous layers' hardness suffers a substantial loss of resistance, in some cases of more than 90%.

Based on the indications of Reviewer 1, this part of the text has been revised and the sentence indicated is no longer present in the paper.

Thank you for your support.

Best Regards
Luca Borro